# The Fifteen Puzzle—A New Approach through Hybridizing Three Heuristics Methods

Dler O. Hasan [1,*], Aso M. Aladdin [1,2], Hardi Sabah Talabani [1], Tarik Ahmed Rashid [3] and Seyedali Mirjalili [4]

1  Department of Applied Computer, College of Medicals and Applied Sciences, Charmo University, Sulaymaniyah, Chamchamal 46023, Iraq
2  Department of Technical Information Systems Engineering, Erbil Technical Engineering College, Erbil Polytechnic University, Erbil 44001, Iraq
3  Computer Science and Engineering Department, University of Kurdistan Hewler, Erbil 44001, Iraq
4  School of Information and Communication Technology, Griffith University, Brisbane, QLD 4111, Australia
*  Correspondence: dler.osman@charmouniversity.org

**Abstract:** The Fifteen Puzzle problem is one of the most classical problems that has captivated mathematics enthusiasts for centuries. This is mainly because of the huge size of the state space with approximately $10^{13}$ states that have to be explored, and several algorithms have been applied to solve the Fifteen Puzzle instances. In this paper, to manage this large state space, the bidirectional A* (BA*) search algorithm with three heuristics, such as Manhattan distance (MD), linear conflict (LC), and walking distance (WD), has been used to solve the Fifteen Puzzle problem. The three mentioned heuristics will be hybridized in a way that can dramatically reduce the number of states generated by the algorithm. Moreover, all these heuristics require only 25 KB of storage, but help the algorithm effectively reduce the number of generated states and expand fewer nodes. Our implementation of the BA* search can significantly reduce the space complexity, and guarantee either optimal or near-optimal solutions.

**Keywords:** Fifteen Puzzle; heuristic search; inadmissible heuristic function; metaheuristic; bidirectional search; unidirectional search

## 1. Introduction

The Fifteen Puzzle is a standard sliding puzzle invented by Samuel Loyd in the 1870s [1] that consists of 15 tiles with one tile missing within a 4 × 4 grid. The fifteen tiles are numbered from 1 to 15. The numbered tiles should initially be ordered randomly. The aim of the game is to slide the tiles that are located next to the space into the space (one at a time) to achieve the numerical order of the tiles from left to right with the blank at the bottom right/top left corner in a minimum time and with minimum moves. Automatically solving the Fifteen Puzzle is very challenging because the state space for the Fifteen Puzzle contains about 16! /2≈$10^{13}$ states [2]. The Fifteen Puzzle contains 16! instances but only half of the instances are solvable [3,4]. Optimal solutions for any solvable instances of the Fifteen Puzzle can take from 0 to 80 moves [5,6]. The two common heuristic search algorithms, A* [7] and iterative deepening A* (IDA*) [8], have been successfully used for computing optimal solutions for the Fifteen Puzzle instances. These algorithms are guided by heuristic functions, which are estimates of the number of moves required to solve any given puzzle configuration.

The most common heuristic functions that have been used to reduce the search space are misplaced tile (MT), MD, LC and pattern databases (PDBs) [9–11]. WD has also been used, but is not common. MT is the number of tiles that are not in their goal positions. MD is the sum of the distance of each tile from its goal position. LC is the sum of two moves for each pair of conflicting tiles that are in their goal row or column positions but

in the wrong order. The WD was developed by [12], and counts the vertical moves and horizontal moves separately, while considering the tiles' conflict with each other. PDBs are heuristics in the form of lookup tables. The two heuristics misplaced tile and Manhattan distance were used with the A* algorithm for optimally solving the Eight Puzzle problem by [13]. The Manhattan distance and linear conflict heuristics were combined and used with the IDA* algorithm for the Fifteen Puzzle by [14]. The walking distance heuristic was developed and used (with the IDA* search) by [12] for the Fifteen Puzzle. To the best of our knowledge, the walking distance heuristic has not been used in any research for the Fifteen Puzzle. The pattern database heuristics were first introduced by [15] and then used by many researchers, and now there are various types of pattern databases [16]. The main drawback of pattern databases is that they require a large amount of memory (several gigabytes for some types of pattern databases) [16,17]. Flener, Korf, and Hanan [18] claimed that an effective heuristic for the Fifteen Puzzle is the 7–8 additive pattern database, but this heuristic requires a lot of storage space and can be memory-intensive at about 575 megabytes.

All the heuristics used to estimate how close a state is to the goal suffer from a number of drawbacks. For example, some of them are not very accurate at estimating the remaining distance to a goal, such as MD, MT, and WD, and the others are accurate but require a lot of storage space, such as PDBs. The main objective of this paper is to combine some heuristics to accurately estimate the cost from the current state to the goal state without generating a lot of states or requiring a large amount of storage space to store the nodes. The contribution of this paper is to hybridize the three heuristics MD, LC, and WD to estimate the number of steps to the goal state. Moreover, to increase the effectiveness of the heuristic function, the MD value is divided by three. We use this heuristic in such a way to significantly reduce the number of generated nodes to solve the puzzle states. Using these heuristic algorithms in this way cannot be guaranteed to give an optimal solution, but they usually find an optimal solution or a solution that is one to six moves away from the optimum and, in some rare cases, more than six moves away from the optimum. For the most difficult states, we run two searches—a forward search from the initial state and a backward search from the end state (goal state), which is called a bidirectional search. This is for the sake of improving the algorithm performance.

This paper is structured as follows. Section 2 is devoted to presenting and discussing the implementation of our BA* algorithm. Section 3 presents and evaluates the three heuristics we use to solve the Fifteen Puzzle problem. Section 4 presents the efficient way of hybridizing the three heuristics for solving the Fifteen Puzzle. Section 5 presents and discusses the results and their comparisons. Section 5.1 compares our implementation of the BA* algorithm with the artificial bee colony (ABC) algorithm in terms of efficiency and inadmissibility. Section 5.2 discusses the comparison between bidirectional A* (BA*) search and unidirectional A* (UA*) search. Section 5.3 describes the experiments performed with our implementation of the BA* algorithm and also compares the results obtained by our algorithm with the results obtained by the IDA* algorithm with MD and LC heuristics. Finally, Section 6 highlights the main conclusion of this study.

## 2. Bidirectional A* Algorithm

The IDA* and A* are the two most popular heuristic search algorithms widely used to solve the Fifteen Puzzle problem. The A* algorithm is one of most well-regarded algorithms in artificial intelligence for finding the shortest path or the smallest number of moves from the initial state to the goal [7]. Despite being complete, this algorithm has some disadvantages that can make it inefficient, especially for complex and large puzzle problems. This is because billions of nodes need to be expanded and generated for the difficult states, and in the A* algorithm all the generated nodes are kept in memory, which can lead to running out of memory, or sometimes finding a solution takes a long time. The IDA* algorithm is a variant of the A* algorithm that can be implemented for solving the Fifteen Puzzle [8]. Due to the fact that IDA* does not store the expanded nodes in memory,

it uses less space and expands nodes faster than the A* algorithm. Even though the IDA* algorithm is more efficient than the A* algorithm, we still use the A* algorithm in this paper for several reasons. First of all, since we use bidirectional search, the A* algorithm is a good choice because it stores all the generated nodes in memory, and this leads to frontier intersections that can be easily tested [9]. Secondly, the A* algorithm with the heuristics that we use only generates a few states, and this does not cause the algorithm to run out of memory. Thirdly, since the A* algorithm retains the generated states in memory, each state is generated once.

Today, there are several variants of the A* algorithm since its performance can be varied depending on several methods. One of the optimization methods is bidirectional search, which is commonly used nowadays [19–22] since it can improve the efficiency of the algorithm in terms of space and time complexity. Another method is heuristic function, which directly affects the algorithm's efficiency, and, therefore, there are several heuristics proposed for estimating the distances to remaining unexplored nodes [20]. Additionally, there are other variants of the A* algorithm in which the heuristic cost is weighted differently, and this has been utilized to speed up the A* algorithm [19,22]. This paper utilized all three mentioned methods to improve the A* algorithm's efficiency.

Algorithm 1 gives the pseudocode for the bidirectional A* (BA*) algorithm. Some notations are used, such as OpenList, ClosedList, and NeighboringState, which denote the states that have been visited but not expanded, the states that have been visited and expanded, and the state that is directly connected to the current state. There are separate copies of these variables for both the forward and backward search, with a subscript (F or B) indicating the direction:

Forward search: $\text{OpenList}_f$, $\text{ClosedList}_f$, and $\text{NeighboringState}_f$, etc.

Backward search: $\text{OpenList}_b$, $\text{ClosedList}_b$, and $\text{NeighboringState}_b$, etc.

The BA* algorithm for each one of the two searches (forward and backward search) needs two lists: a closed list, which is used for storing all the puzzle states that have been visited and expanded, and an open list, which is used for storing the puzzle states that have been visited but not expanded. At each step, the heuristic value and the depth cost of the current state is determined. Then, the states inside the open list are sorted according to the heuristic values in increasing order. At every step, the head of the open list, which has the lowest evaluation function value (which is the heuristic value plus the path cost), is removed from the open list and then checked to see whether it is the goal state (start state for backward search) or not. If the head state is the goal state (start state for backward search), the algorithm reconstructs the path to the goal (to the start for backward search). If the head state is not the goal (is not the start for backward search), it is checked to see if it is in the closed list of the opposite search direction, and if it is there, the algorithm reconstructs the solution path from the two searches. When the goal (start for backward search) is not found, the head state is expanded (all the valid moves are specified) and it is placed on the closed list. Then, all the successors of the head state which are not already on the closed list are stored in the open list. As is shown in Algorithm 1, the forward search starts first and continues until 75,000 states are expanded, but after the first step of the cycle, the forward search continues until 15,000 states are expanded. If the solution path from the start state to the goal state is not found while generating this number of states, the forward search stops, and the backward search starts. The backward search continues until 75,000 states are expanded (until 15,000 states are expanded after the first step of the cycle). If, during that period, the solution path from the goal state to the start state is not found, the backward search stops, and the forward search starts again. This process will continue until the solution path is found.

---

**Algorithm 1. BA\* algorithm pseudocode**

---

**function** BA\* (*StartState*, *GoalState*)
　　**Initialise:**
　　　　*Iterator$_f$* to control the loop
　　　　*OpenList$_f$* to store the states to be traversed
　　　　*ClosedList$_f$* to store already traversed states
　　　　*OpenList$_b$* to store the states to be traversed
　　　　*ClosedList$_b$* to store already traversed states
　　**if** *Iterator$_f$* = 0 **then**
　　　　set depth cost of *StartState* (g(s) in Equation (2)) to zero
　　　　calculate HH value from *StartState* to *GoalState*. Equation (3)
　　　　calculate evaluation function for *StartState*. Equation (2)
　　　　add *StartState* into *OpenList$_f$* and *ClosedList$_f$*
　　**while** *OpenList$_f$* is not empty **do**
　　　　*CurrentState$_f$* is state with lowest evaluation function value (Equation (2)) in *OpenList$_f$*
　　　　remove *CurrentState$_f$* from *OpenList$_f$*
　　　　**if** *CurrentState$_f$* is *GoalState* **then**
　　　　　　reconstruct the solution path from *StartState* to *CurrentState$_f$*, and terminates the loop
　　　　**for each** *NeighboringState$_f$* of *CurrentState$_f$* **do**
　　　　　　**if** *NeighboringState$_f$* is not in *ClosedList$_f$* **then**
　　　　　　　depth cost of *NeighboringState$_f$* is equal to the depth cost of *CurrentState$_f$* plus one
　　　　　　　calculate HH value from *NeighboringState$_f$* to *GoalState*. Equation (3)
　　　　　　　calculate evaluation function for *NeighboringState$_f$*. Equation (2)
　　　　　　　add *NeighboringState$_f$* into *ClosedList$_f$*
　　　　　　　add *NeighboringState$_f$* into *OpenList$_f$*
　　　　　　　**if** *NeighboringState$_f$* is in *ClosedList$_b$* **then**
　　　　　　　　reconstruct the solution path from the two searches: from *StartState*
　　　　　　　　to *NeighboringState$_f$*
　　　　　　　　and from *NeighboringState$_f$* to *GoalState*, and terminates the loop
　　　　　　increase *Iterator$_f$* by 1
　　　　**if** *Iterator$_f$* mod 15000 is equal to 0 after the first step of the cycle or *Iterator$_f$* mod 75000
　　　　is equal to 0 **then**
　　　　　　->Expand in the backward direction, analogously

---

To further explain Algorithm 1, Figure 1 illustrates how the two searches work. The puzzle state used in Figure 1 is Korf's 15-puzzle problem instance #82 [8]. At the beginning, the first search (forward search) starts from the initial state and the evaluation function value (Equation (2)) for it is calculated, which is the summation of the heuristic value (Equation (3)) and the depth cost (*g*). After expanding the initial state, two new puzzle states are generated at depth one, the evaluation function value for each of them is calculated, and the state with the minimum evaluation function value is expanded (as shown in Figure 1, the state with the evaluation function value 63 is expanded). At each step, the puzzle state is checked to see whether it is the goal state or not or, or whether it is in the closed list of the opposite search direction. If the goal state is not found after expanding 75,000 states, the backward search starts from the goal state. The same process as the forward search is conducted. As illustrated in Figure 1, at depth 50, the backward search meets a puzzle state which was already generated by the forward search at depth 18. Then, the algorithm reconstructs the path from the two searches, which is 68 moves.

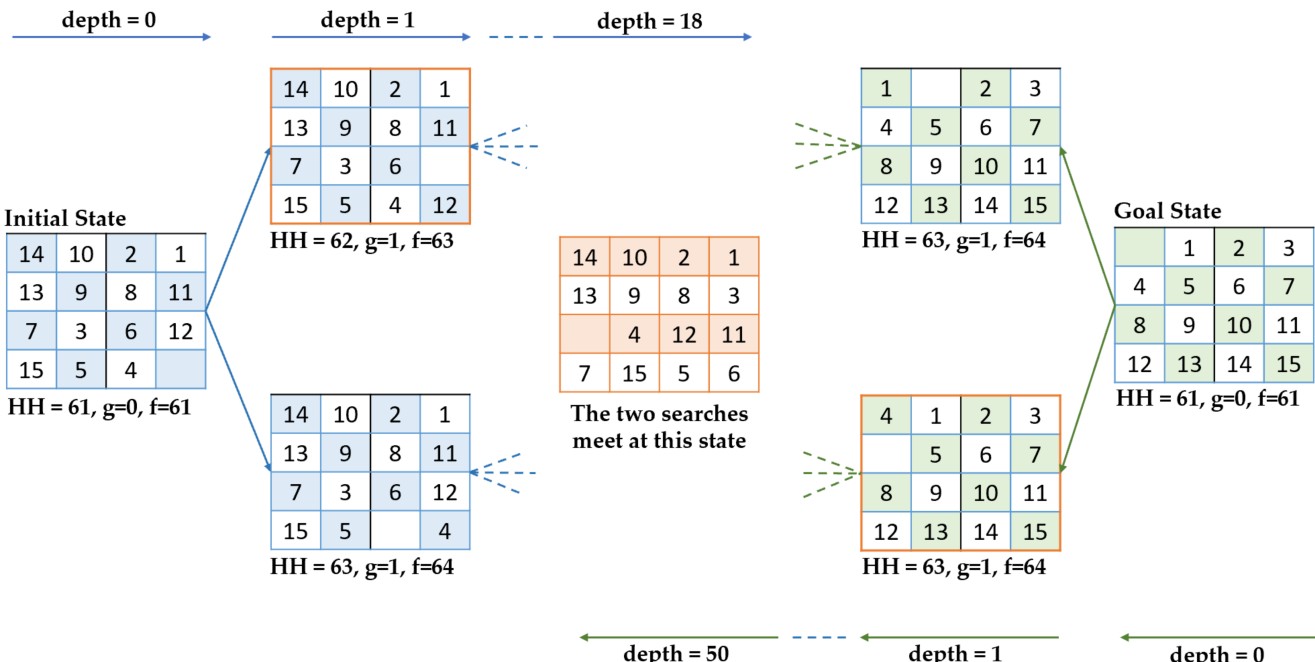

**Figure 1.** A numerical example to illustrate the BA* algorithm.

## 3. Heuristic Functions

A heuristic is an informed guess to choose the next node to visit when exploring a search space. A heuristic can lead the algorithm to a solution or fail to reach the goal. The three heuristics which are used in this paper are Manhattan distance, walking distance, and linear conflict. Figure 2a shows an arbitrary start state of the Fifteen Puzzle and (b) shows the goal state of the Fifteen Puzzle. The tiles are denoted by $t_i$ and the blank by $t_0$. The sequences are $<t_1, t_4, t_2, t_3, t_{13}, t_6, t_7, t_8, t_5, t_{10}, t_{11}, t_0, t_9, t_{14}, t_{15}, t_{12}>$ for the start state and $<t_1, t_2, t_3, t_4, t_5, t_6, t_7, t_8, t_9, t_{10}, t_{11}, t_{12}, t_{13}, t_{14}, t_{15}, t_0>$ for the goal state, as shown in Figure 2.

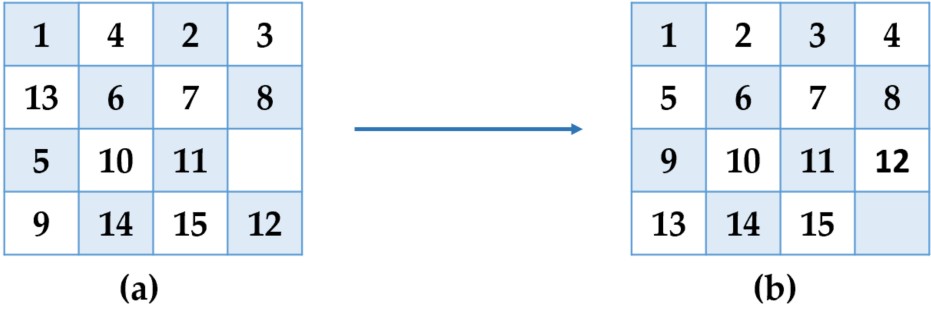

**Figure 2.** Fifteen Puzzle (**a**) start state, (**b**) goal state.

The Manhattan distance of a puzzle is the sum of the horizontal and vertical distance of each tile (except the blank tile) from its goal position [8]. For the initial state of the Fifteen Puzzle shown in Figure 2, only the tiles $t_4, t_2, t_3, t_{13}, t_5, t_9$, and $t_{12}$ are not in their goal positions, and they are away from their goal positions by 2, 1, 1, 2, 1, 1, and 1, respectively. Therefore, the heuristic function value is 9 (2 + 1 + 1 + 2 + 1 + 1 + 1). This means that the current state needs at least 9 moves to reach the goal. Manhattan distance is admissible because it never overestimates the number of moves to the goal and each tile must at least be moved from its current position to its goal position, and only vertical and horizontal movement is allowed. Therefore, the Manhattan distance value of any state is less than

or equal to the number of moves that the state needs to reach the goal. The Manhattan distance of a tile in a puzzle can be found using Equation (1) (*s* is the current state) [17]:

$$h(s) = \sum_{i=1}^{n} (|x_i(s) - \overline{x_i}| + |y_i(s) - \overline{y_i}|) \tag{1}$$

LC, which is used to enhance the effectiveness of the Manhattan distance, adds two additional moves to the Manhattan distance for each pair of conflicting tiles that would have to be swapped to reach the goal state. Two tiles, $t_i$ and tj, are in a linear conflict if both tiles are positioned in their goal row or column but in the wrong order, or, in other words, they are reversed relative to their goal location [14]. For example, in Figure 2, the tile $t_4$ conflicts with tiles $t_2$ and $t_3$, and by changing the row of tile $t_4$ we can eliminate these conflicts. The tile $t_{13}$ conflicts with tiles $t_5$ and $t_9$ because they are in the correct column but inverse order. In this case, $t_9$ must move one place to the right to let the others pass by and then move back to its column position. These four moves are not counted in the Manhattan distance. Therefore, two additional moves are added to the Manhattan distance for each pair of conflicting tiles and the heuristic evaluation function remains admissible.

So far, the total cost function for the initial state in Figure 2 is equal to 13 (9 for the Manhattan distance, 4 for the linear conflict), while the optimal solution for the initial state is 29 moves. Therefore, using these two heuristics cannot make the algorithm efficient, especially for complex and large puzzle problems, and finding the solution takes a long time. This is because Manhattan distance does not capture the conflicts and interactions between the tiles, and this leads to heavily underestimating the actual optimal solution cost in almost all the problem instances of the Fifteen Puzzle [23], and linear conflict only adds two moves for every two tiles which are positioned in the correct row/column but inverted. Walking distance counts the vertical moves and horizontal moves separately while considering the tiles' conflict with each other [12]. According to the goal state in Figure 2, all four tiles ($t_1$, $t_4$, $t_2$, and $t_3$) in the first row of the initial state are from the first row of the goal state and no tiles are from the other rows of the goal state. The same approach is used for the other rows, as is shown in Table 1.

**Table 1.** Walking distance calculation.

| No. of Rows | Number of Tiles from 1st Row | Number of Tiles from 2nd Row | Number of Tiles from 3rd Row | Number of Tiles from 4th Row | Blank Tile |
|---|---|---|---|---|---|
| **1st row** | 4 | 0 | 0 | 0 | |
| **2nd row** | 0 | 3 | 0 | 1 | |
| **3rd row** | 0 | 1 | 2 | 0 | ← here |
| **4th row** | 0 | 0 | 2 | 2 | |

To calculate the horizontal walking distance, we can only swap the blank tile with any single tile from the row above or below, and the order of the tiles in each row is irrelevant. We keep swapping until all the tiles are in their goal rows. The minimum number of moves needed to take all the tiles to their goal row positions is the horizontal walking distance. We can apply the same procedure to calculate the vertical walking distance by taking all the tiles to their goal column positions with the minimum number of moves, and each tile can only be taken into a column adjacent to the column containing the blank tile and swap places with it. The order of the tiles in each column is irrelevant. The total walking distance is the sum of the number of horizontal and vertical moves. To further explain Table 1, Figure 3 illustrates how WD can be calculated manually step by step for the initial state in Figure 2. Two 4 × 4 tables are needed, one for computing the horizontal WD value and another one for computing the vertical WD value.

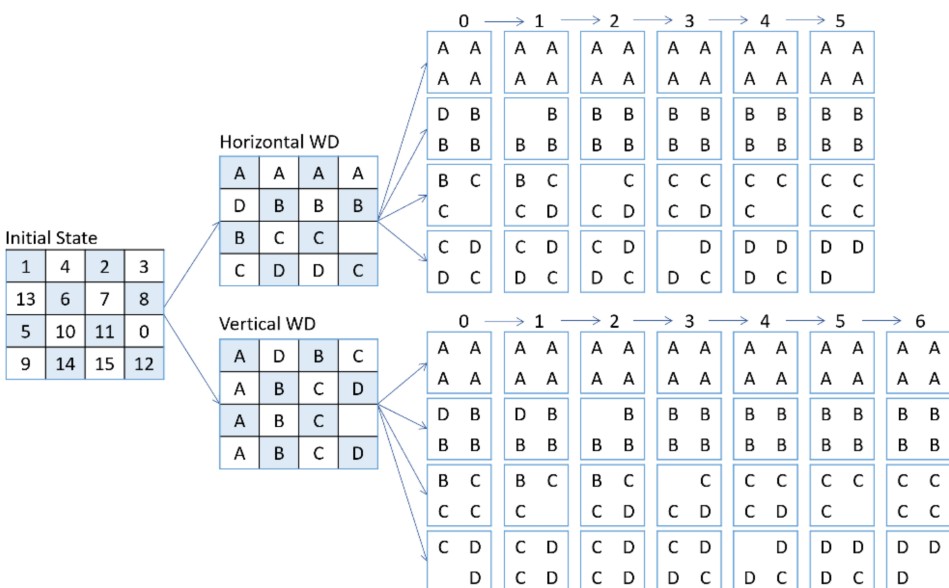

**Figure 3.** Step by step walking distance calculation.

The table of the horizontal WD in Figure 3 has four 'A' elements in the first row, which means that all the tiles $t_1$, $t_4$, $t_2$, and $t_3$ are from the first row of the goal state. It has one 'D' element and three 'B' elements in the second row, which is because $t_{13}$ is from the fourth row of the goal, and $t_6$, $t_7$, and $t_8$ are from the second row of the goal. It has one 'B' element and two 'C' elements with a blank ($t_0$) element in the third row, which is because $t_5$ is from the second row of the goal, and $t_{10}$ and $t_{11}$ are from the third row of the goal. It has two 'C' elements and two 'D' elements in the fourth row, which is because $t_9$ and $t_{12}$ are from the third row of the goal, and $t_{14}$ and $t_{15}$ are from the fourth row of the goal. As is shown in Figure 3, only five steps are needed to take all the tiles to their goal row position, which is the horizontal WD value for the initial state in Figure 2. The same procedure is used for building the table of the vertical WD and calculating the vertical WD value, except that when we build the table of vertical WD, we must specify which column each tile's goal position is in. As can be seen in Figure 3, to take all the tiles to their column position, six steps are needed, which is the vertical WD value for the initial state in Figure 2. The total walking distance is the sum of the number of horizontal and vertical moves, which is 11 steps.

Since walking distance cannot be easily computed at runtime, we can precompute all these values and store them in the database, because if we do not precompute them, this heuristic can slow the search down significantly. Instead of fully calculating the walking distance during the search, a breadth-first search (BFS) can be executed backward from the goal state to obtain all the distinct tables for all the Fifteen Puzzle configurations (all possible configurations of the tiles), which are only 24,964 patterns, and store them in the database to speed up the search. The size of the database is relatively small at about 25 KB. The same database is used for calculating the number of horizontal and vertical moves. The maximum walking distance value is 70 (such as $t_0$, $t_{15}$, $t_{14}$, $t_{13}$, $t_{12}$, $t_{11}$, $t_{10}$, $t_9$, $t_8$, $t_7$, $t_6$, $t_5$, $t_4$, $t_3$, $t_2$, $t_1$), with 35 moves each for horizontal and vertical moves. WD is more accurate and efficient than Manhattan distance because the WD value is always greater than the MD value, as is illustrated in Figure 4.

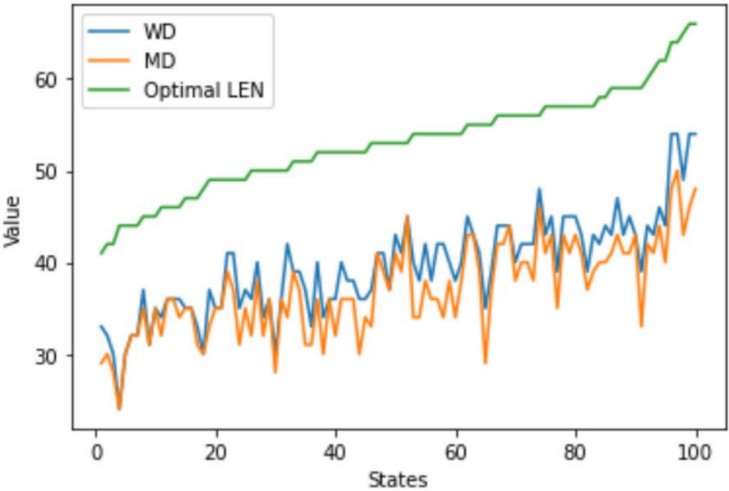

**Figure 4.** MD, WD, and optimal value for Korf's 100 instances.

Figure 4 shows the MD, WD, and optimal values for Korf's 100 instances [8] after sorting the instances by optimal value. In all of them, the WD value is greater than the MD value, and Table 2 shows that the total WD value for all the 100 instances is greater than the total MD value. Table 2 also shows the minimal total cost (optimal solution) and total LC values for all the 100 instances.

**Table 2.** Total WD, MD, LC, and optimal solution lengths for Korf's 100 instances of the Fifteen Puzzle.

| Problems | Total WD | Total MD | Total LC | Total Optimal |
|---|---|---|---|---|
| **Korf's 100 instances** | 3957 | 3705 | 212 | 5307 |

The walking distance can also be enhanced by the linear conflict, because WD does not count the two moves which are determined by linear conflict for each pair of conflicting tiles. As shown in Figure 3, when calculating the horizontal or vertical WD values when we have two tiles in linear conflict, the first tile can slide to the row above or below, if that row contains a blank, without moving the second tile, and for the second tile this is also correct. For example, Figure 5 zooms in and shows a part of Figure 3 where the tile $t_{13}$ (D) that conflicts with the tile $t_5$ (B) can slide to the third row without moving the tile $t_5$ (B).

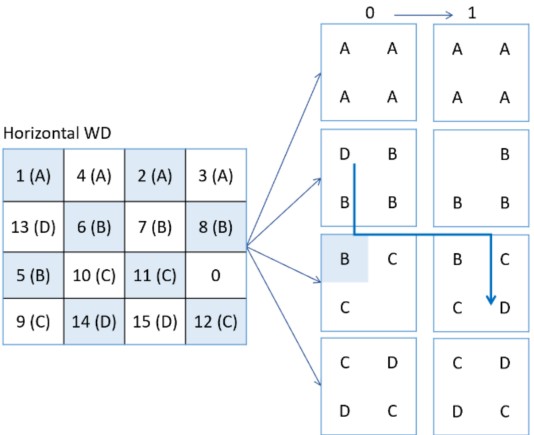

**Figure 5.** WD does not capture the LC heuristic.

We have built the walking distance lookup table for both goal states with a blank at the bottom right and top left corner because the two different goal states have been used in many types of research and we also use these two different goal states in this paper.

## 4. Hybridized Heuristic Functions

Since we have no perfect heuristic function (exact distance function) to give us the exact number of moves needed to solve all the Fifteen Puzzle instances and each heuristic has its way to calculate the distance between the current state to the goal state, it is desirable to combine multiple heuristics which can complete each other to estimate the solution cost. Combining several heuristics is generally the best way to more accurately estimate the cost of reaching the goal, but it is challenging [24–26]. Multiple heuristics have been used in different ways. The most common way to use multiple heuristics is to combine different heuristics and use their maximum value. Holte et al. [16] showed that taking the maximum heuristic value from several heuristics can lead to reduced node generation and result in improved performance of the search. When two or more admissible heuristics are combined, taking their maximum, that is, by defining (hmax(s) = max(h1(s), h2(s)). ), is also admissible [16,27].

Another way to use multiple heuristics is cost partitioning or cost splitting, which has been used by many researchers [28–30] and is a technique to add multiple heuristic values in an admissible way with operator cost partitioning by distributing the cost of each operator among them. This technique has a drawback in finding good cost partitioning [29]. Korf and Taylor [31] took advantage of several heuristics including Manhattan distance, linear conflict, last moves, and corner tile to improve the accuracy of the heuristic evaluation function and result in improved search performance of the IDA* search. In addition, they used the heuristics in a way that keeps the them admissible; for example, when the same tile is involved in a corner tile and linear conflict, the extra moves are added only once. Therefore, whenever we combine multiple heuristics and we want to find the optimal solution, we must be sure that the actual distance for any tile is not calculated more than once. Those heuristics are not complex, and it is easy to check which tiles are involved in multiple heuristics. Manhattan distance and walking distance, the two heuristics that we use in this paper, are complex and it is not easy to check which tiles' actual distance to their goal position is counted by the two heuristics.

Each heuristic has its strengths and weaknesses. Therefore, we must determine the weaknesses and strengths of the heuristics when we want to combine multiple to create a more accurate heuristic function. The main drawback of Manhattan distance is measuring each tile's distance to its goal position without considering the interference from any other tiles [18]. For example, according to the MD, the tiles $t_6$, $t_7$, $t_8$, $t_{10}$, $t_{11}$, $t_{14}$, and $t_{15}$ in the initial state shown in Figure 3 need zero moves to reach their goal positions since they are already in them. This estimation is not correct because it is not possible to take the tiles $t_4$, $t_2$, $t_3$ $t_{13}$, $t_5$, and $t_9$ to their goal positions without moving some of the tiles $t_6$, $t_7$, $t_8$, $t_{10}$, $t_{11}$, $t_{14}$, and $t_{15}$. On the other hand, WD considers interactions between tiles and it calculates the distance of the tiles to their goal positions like MD. As illustrated in Figure 6, which is a part of Figure 3, one of the tiles $t_7$, $t_{11}$, and $t_{15}$ makes two moves while calculating the WD value, which proves that WD is more efficient than MD.

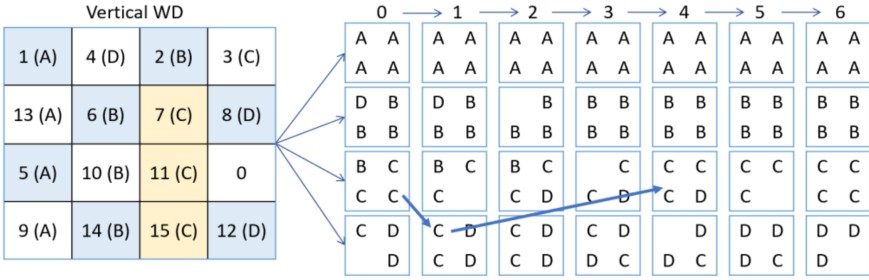

**Figure 6.** WD considers the interference of tiles with each other.

WD is not exactly equal to MD plus the interference of tiles with each other. It seems that the WD heuristic considers interactions between tiles and the distance of each tile to its goal position, but we think it finds the interaction of tiles to be more significant than their

distances to their end positions, because there are many Fifteen Puzzle problem instances that have the same MD and WD values. For instance, 23 out of Korf's 100 instances have the same WD and MD values despite a lot of interactions between their tiles. Additionally, for the instance <$t_{15}$, $t_4$, $t_7$, $t_{11}$, $t_5$, $t_8$, $t_0$, $t_3$, $t_{14}$, $t_2$, $t_{12}$, $t_{13}$, $t_1$, $t_6$, $t_{10}$, $t_9$>, the WD and MD are both 35 even though there are a lot of interactions between the tiles. If the WD is equal to the exact MD plus the conflicts between the tiles, the WD value for that Fifteen Puzzle instance must be greater than the MD value, not equal to it. Therefore, this proves that the WD is not equal to the exact MD plus the conflicts between the tiles for the Fifteen Puzzle problem instances. In spite of this, WD works somewhat similarly to but not exactly the same as MD, since WD takes each tile to its goal column–row position when calculating the horizontal and vertical values for a puzzle instance, as is illustrated in Figure 3. In general, WD is more efficient and better than MD because WD is never less than MD, as is illustrated in Figure 4. Because of this, we use WD and LC as the main heuristics together with MD as a helping heuristic to assist the main heuristics. Since we use MD as a helping heuristic, the MD value is divided by 3. In this way, the MD value is reduced to a number when it is added to the main heuristics values, so the result will be close to the optimal solution length.

As explained before, WD mainly considers the interactions between the tiles, and it calculates the distance of the tiles to their actual positions in a way similar to MD. Therefore, to compensate, the calculating tiles' distance to their goal position MD is used but not the whole MD value. The MD value is divided by a number (which is three) so that the summation of WD, LC and MD/3 will be close to the optimal solution length. For example, if we sum the total WD value (3957), LC value (212), and MD value divided by three (3705/3), as is shown in Table 2, the result will be 5404, and this result is near to the total optimal solution value 5307 for Korf's standard 100 random Fifteen Puzzle instances. Furthermore, this total overestimation is very small and it does not have a great impact on the results of BA* with HH, as the implementation of our algorithm finds an optimal solution or a solution that is near to the optimum and to reach the goal for each instance, a small number of states are generated. Because of this, we calculate the heuristic function in the evaluation function (Equation (2) [1]), as shown in Equation (3), named hybridizing heuristic (HH). To find the shortest path, the A* algorithm uses the evaluation function as it is shown in Equation (2), which is equal to the $g(s)$, the depth cost from the start state to the current state, plus the $h(s)$, the heuristic that estimates the distance from the current state to the goal state. The A* algorithm guarantees the optimal solution if the heuristic function is admissible.

$$f(s) = g(s) + h(s) \tag{2}$$

$$h(s) = \frac{md(\mathbf{s})}{3} + wd(s) + lc(s) \tag{3}$$

## 5. Results and Discussions

In this section, to evaluate the efficiency and performance of our implementation of the BA* algorithm, we make some comparisons. Firstly, BA* with HH is compared with the ABC algorithm in terms of admissibility. Secondly, in terms of directionality, BA* and UA* are compared to show that the bidirectional search is more efficient than the unidirectional search, especially when there is a guarantee that the two bidirectional searches do not pass by each other without intersecting. Finally, the BA* search with HH is run on Korf's 100 instances, along with a comparison with the IDA* search.

### 5.1. Inadmissible Heuristics

An algorithm can guarantee finding the shortest path or the smallest number of moves from the initial state to the goal only if the heuristic function never overestimates the actual path cost from the current state to the goal state, which we call an admissible heuristic [32]. Due to the reason that finding the optimal solution for the Fifteen Puzzle is too expensive and requires searching through a very large number of paths and generating

a large number of nodes [33], many types of research have been conducted to obtain near-optimal solutions instead of exact optimal solutions [34,35]. Thayer, Dionne, and Ruml [36] state that, to reduce the solving time, a near-optimal solution is a practical alternative. To reduce the number of generated nodes, we incorporated aspects from the three heuristics to create a better one and the heuristic function in the evaluation function (Equation (2) [1]) is calculated as shown in Equation (3). As shown in Equation (3), three heuristics are combined to estimate the cost from a given state (node) to the goal state. The value of the Manhattan distance is divided by three because calculating in that way leads to fewer nodes to be generated during the search. Because of the previous reasons, the algorithm heuristic cannot guarantee finding the goal with the smallest number of moves, but this brings some advantages. Firstly, a lesser number of nodes are generated and it can very quickly find the goal. Secondly, the result is very close to the optimal solution. Since a bidirectional search has been used to find the path from the initial state to the goal state, the three heuristics have been used in both directions (search).

However, our implementation of the BA* algorithm with the three heuristics does not find optimal solutions for most of the Fifteen Puzzle instances; the difference between the solution length found by BA* and the optimal solution for each puzzle instance does not increase when the puzzle instance requires more moves to optimally reach the goal. Nowadays, metaheuristic optimization algorithms are widely used for solving complex problems [37–39]. One of the algorithms that has recently been used to obtain non-optimal solutions to the Fifteen Puzzle problems is a metaheuristic algorithm: artificial bee colony (ABC) [40]. Here, the BA* algorithm with HH is compared with the ABC algorithm to show that the obtained results of BA* are sufficiently accurate and much nearer to the optimal results. To increase the effectiveness and performance of the heuristic function of the ABC algorithm, three heuristics, PDB, MD, and LC, were combined. The ABC algorithm was run on 25 randomly generated solvable instances of the Fifteen Puzzle, but the algorithm did not produce an optimal solution for any of them; it provided solutions that are far from the optimum [40]. Tuncer [40] argued that the results produced by the ABC algorithm are acceptable even though the solution lengths are far from the optimal solution lengths. Furthermore, the difference between the solution costs obtained by the ABC algorithm and the optimal solutions for most of the puzzle instances increases when the puzzle instances require more moves to optimally reach the goal. For example, according to Table 4, the solution cost obtained by the ABC algorithm for the first nine puzzle instances that need fewer steps to optimally reach the goal is near to the optima, while the rest of the puzzle instances are very far from the optima. This is because these instances need more steps to optimally reach the goal. According to this example, the difference between the number of moves obtained by the ABC algorithm and the optimal solution will be big, especially for those states that require 80 moves to reach the goal. On the other hand, an important point about our implementation of the BA* algorithm is that the solution lengths for almost all the Fifteen Puzzle instances are zero to six moves away from the optimal solution lengths, even for the difficult states, as is shown in Tables 3 and 4.

The BA* algorithm with HH was run on the same 25 initial states, and the results obtained by the BA* algorithm are very near to the optimal solutions compared to the results obtained by the ABC algorithm. For example, Table 4 shows that the average number of moves in the solutions that are obtained by the ABC algorithm is 58.76, while the average number of moves in the solutions that are obtained by the BA* algorithm is 50.4. In addition, the average number of moves in the solutions found by BA* is only 1.92 away from the average cost of the optimum solution, which is 48.48, while the average number of moves in the solutions found by ABC is 10.28 away from the average cost of the optimum solution. Figure 7 illustrates the results of the 25 states presented in Table 4 obtained by the ABC and BA* algorithm.

**Table 3.** Comparison of BA* search and UA* search for the 28 difficult Fifteen Puzzle instances requiring 80 moves.

| NO | INITIAL STATE | Optimal LEN | LEN (UA*) | Generated States (UA*) | LEN (BA*) | Generated States (BA*) |
|---|---|---|---|---|---|---|
| 1 | 15 14 8 12 10 11 9 13 2 6 5 1 3 7 4 0 | 80 | Memory ran out | | 88 | 187,592 |
| 2 | 15 11 13 12 14 10 8 9 7 2 5 1 3 6 4 0 | 80 | 84 | 138,505 | 84 | 138,505 |
| 3 | 15 11 13 12 14 10 8 9 2 6 5 1 3 7 4 0 | 80 | 82 | 1,605,359 | 86 | 367,391 |
| 4 | 15 11 9 12 14 10 13 8 6 7 5 1 3 2 4 0 | 80 | 82 | 771,924 | 86 | 420,441 |
| 5 | 15 11 9 12 14 10 13 8 2 6 5 1 3 7 4 0 | 80 | 84 | 1,207,604 | 86 | 199,905 |
| 6 | 15 11 8 12 14 10 13 9 2 7 5 1 3 6 4 0 | 80 | 82 | 809,360 | 82 | 185,126 |
| 7 | 15 11 9 12 14 10 8 13 6 2 5 1 3 7 4 0 | 80 | Memory ran out | | 86 | 219,470 |
| 8 | 15 11 8 12 14 10 9 13 2 6 5 1 3 7 4 0 | 80 | 84 | 2,565,243 | 86 | 200,926 |
| 9 | 15 11 8 12 14 10 9 13 2 6 4 5 3 7 1 0 | 80 | 84 | 751,072 | 84 | 190,731 |
| 10 | 15 14 13 12 10 11 8 9 2 6 5 1 3 7 4 0 | 80 | 82 | 1,137,335 | 84 | 205,344 |
| 11 | 15 11 13 12 14 10 9 5 2 6 8 1 3 7 4 0 | 80 | 82 | 1,933,020 | 86 | 530,773 |
| 12 | 0 12 9 13 15 11 10 14 3 7 2 5 4 8 6 1 | 80 | Memory ran out | | 88 | 186,644 |
| 13 | 0 12 10 13 15 11 14 9 3 7 2 5 4 8 6 1 | 80 | 84 | 2,096,287 | 84 | 207,896 |
| 14 | 0 11 9 13 12 15 10 14 3 7 6 2 4 8 5 1 | 80 | 84 | 949,297 | 84 | 198656 |
| 15 | 0 15 9 13 11 12 10 14 3 7 6 2 4 8 5 1 | 80 | 84 | 734,711 | 84 | 167,455 |
| 16 | 0 12 9 13 15 11 10 14 3 7 6 2 4 8 5 1 | 80 | Memory ran out | | 86 | 256,899 |
| 17 | 0 12 14 13 15 11 9 10 3 7 6 2 4 8 5 1 | 80 | 84 | 917,307 | 86 | 205,555 |
| 18 | 0 12 10 13 15 11 14 9 3 7 6 2 4 8 5 1 | 80 | 82 | 1,623,362 | 86 | 341,405 |
| 19 | 0 12 11 13 15 14 10 9 3 7 6 2 4 8 5 1 | 80 | Memory ran out | | 86 | 520,393 |
| 20 | 0 12 10 13 15 11 9 14 7 3 6 2 4 8 5 1 | 80 | 82 | 764,029 | 82 | 199,908 |
| 21 | 0 12 9 13 15 11 14 10 3 8 6 2 4 7 5 1 | 80 | Memory ran out | | 86 | 213,147 |
| 22 | 0 12 9 13 15 11 10 14 8 3 6 2 4 7 5 1 | 80 | 84 | 998,668 | 86 | 205,473 |
| 23 | 0 12 14 13 15 11 9 10 8 3 6 2 4 7 5 1 | 80 | 84 | 1,372,770 | 86 | 416,315 |
| 24 | 0 12 9 13 15 11 10 14 7 8 6 2 4 3 5 1 | 80 | 82 | 1,205,808 | 86 | 213,283 |
| 25 | 0 12 10 13 15 11 14 9 7 8 6 2 4 3 5 1 | 80 | 84 | 105,242 | 84 | 105,242 |
| 26 | 0 12 9 13 15 8 10 14 11 7 6 2 4 3 5 1 | 80 | 82 | 2,259,670 | 86 | 534,581 |
| 27 | 0 12 9 13 15 11 10 14 3 7 5 6 4 8 2 1 | 80 | Memory ran out | | 88 | 160,899 |
| 28 | 0 12 9 13 15 11 10 14 7 8 5 6 4 3 2 1 | 80 | 84 | 2,358,160 | 84 | 209,711 |
| | **Average** | **80** | **83.1** | **1,252,606** | **85.4** | **256,774** |

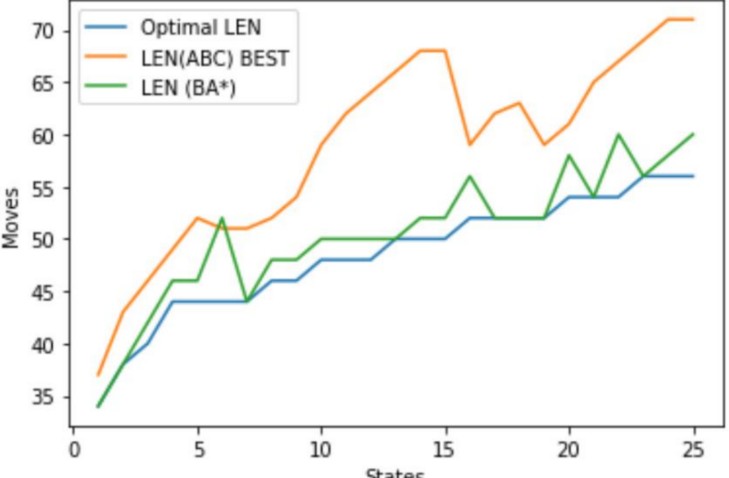

**Figure 7.** Results of 25 Fifteen Puzzle states for ABC and BA* algorithms.

### 5.2. Bidirectional and Unidirectional Search

In a bidirectional search, two separate searches are sequentially or simultaneously run. One search, called the forward search, is normal, starting from the initial state and moving toward the goal state, and the other search, called the backward search, starts from the goal state and moves toward the initial state. The search process terminates once the two searches meet at a common node in the middle and the algorithm constructs a single path

that extends from the initial state to the goal state [41–44]. Pohl [42] was the first person who introduced and implemented a bidirectional heuristic search algorithm with the name bidirectional heuristic path algorithm (BHPA) and he showed that a bidirectional search is more efficient than a unidirectional search. BHPA did not work as expected, since after the search frontiers meet, both directions of searches pass through the opposing frontier to ensure optimality of the solution, and this leads to the same node being expanded by the two searches. To resolve this issue, Kwa [45] created a bidirectional staged BS* heuristic search algorithm, which is derived from Pohl's BHPA algorithm, to avoid the re-expansion of a state that has already been expanded in the opposite search. These days, there are several types of research that prove that a bidirectional search is very efficient at solving various problems [46–50].

As shown in Algorithm 1, we implemented a bidirectional search as follows: two sequential processes are run, one branching from the start state, the other branching from the goal state. The first search, "forward search", starts from the initial state and will continue until 75,000 nodes are expanded. If the goal state is not reached, the second search, "backward search", is initiated from the goal state towards the initial state and this search will continue until it expands 75,000 nodes. If the goal is not found, the backward search stops, and the forward search is performed again. The search process will continue to cycle until both directions meet or the solution is found. During the search, whenever a state is generated by one of the two searches, the algorithm checks if the state has already been generated by the opposite search, and, if it has, it reconstructs a solution path from the two searches. Korf and Schultze [51] were able to compute the number of unique states at each depth of the Fifteen Puzzle. According to [51], the number of generated nodes at each depth gradually increases from depth 0 to depth 53, then the number of generated nodes at each depth starts to gradually decrease from depth 54 to 80. Based on this, the bidirectional search may not be very effective, because the number of generated nodes at depth 53 decreases in both directions and it can be difficult for both searches to meet in the middle. Therefore, one of the problems for the bidirectional search is that the two searches may not meet or may pass by each other without intersecting, but the A* algorithm retains all the visited nodes in the memory, which ensures that the two searches meet and frontier intersections can be easily tested [9]. Furthermore, there can be more than one optimal solution or non-optimal solution for the Fifteen Puzzle instances that can help the two searches not pass by each other without intersecting [52]. Additionally, the bidirectional search is very useful when the problem has not had many goals.

Our implementation of the BA* search can reduce the number of generated states because we use a priority queue to store the estimated costs of states (nodes), and the state from anywhere in the entire queue (not at a specific level) with the lowest evaluation function value (the heuristic value plus the path cost) is always selected to expand. Therefore, the algorithm visits the states in order of their costs, not level by level, which results in speeding up the search. Our implementation of the BA* search can find optimal or near-optimal solutions to even the difficult states and with a small fraction of states expanded (and stored) compared to the unidirectional A* (UA*) search. Table 3 shows that the bidirectional search is more efficient than the unidirectional search concerning generated nodes. In Table 3, we run the BA* and UA* search on 28 different states that require 80 moves. The goal state with a blank tile in the top left corner is used for the first 11 instances, but the goal state with a blank tile in the bottom right corner is used for the remaining 17 instances. The first nine instances were presented by [53], instances 10 and 11 were presented by [54], and the last 17 states were found by [55]. According to Table 3, the BA* search is more efficient than the UA* search in terms of node expansion, and, for seven of the states, UA* is unable to find a solution path and it runs out of memory before finding a solution. Even though the average solution cost obtained by UA* is less than the average solution cost obtained by BA*, the difference is not significant; it is only 2.3. Additionally, the average number of states generated by the BA* search is significantly less

than those generated by the UA* search, even though the number of states generated by the UA* search for seven states has not been counted due to running out of memory.

**Table 4.** Comparison of results between BA* algorithm and ABC algorithm.

| NO | INITIAL STATE | Optimal LEN | LEN (ABC) BEST | LEN (BA*) |
|----|---------------|-------------|----------------|-----------|
| 1 | 1 5 2 7 10 14 11 6 15 12 9 3 13 0 8 4 | 34 | 37 | 34 |
| 2 | 5 6 10 7 1 3 11 8 13 4 15 9 14 0 2 12 | 38 | 43 | 38 |
| 3 | 1 11 6 2 10 13 15 5 3 12 0 4 9 7 14 8 | 40 | 46 | 42 |
| 4 | 6 5 2 7 13 0 10 12 4 1 3 14 9 11 15 8 | 44 | 49 | 46 |
| 5 | 4 3 10 7 6 0 1 2 12 15 5 14 9 13 8 11 | 44 | 52 | 46 |
| 6 | 4 10 3 2 1 0 7 8 9 6 13 15 14 12 11 5 | 44 | 51 | 52 |
| 7 | 3 4 11 2 9 1 14 15 7 6 0 8 5 13 12 10 | 44 | 51 | 44 |
| 8 | 3 10 2 5 15 6 13 4 0 11 1 7 9 12 8 14 | 46 | 52 | 48 |
| 9 | 9 4 0 3 14 7 5 12 15 2 13 6 10 1 8 11 | 46 | 54 | 48 |
| 10 | 7 1 12 10 6 11 15 4 0 2 5 14 3 13 8 9 | 48 | 59 | 50 |
| 11 | 1 13 5 7 14 9 10 12 11 8 2 15 6 0 4 3 | 48 | 62 | 50 |
| 12 | 13 9 5 12 10 2 4 11 3 8 0 7 1 14 6 15 | 48 | 64 | 50 |
| 13 | 2 13 6 1 14 5 11 0 12 4 8 10 9 3 15 7 | 50 | 66 | 50 |
| 14 | 11 3 12 9 2 8 10 14 0 7 15 13 1 6 5 4 | 50 | 68 | 52 |
| 15 | 7 6 15 12 14 1 13 3 0 9 8 4 2 11 5 10 | 50 | 68 | 52 |
| 16 | 5 8 13 15 14 0 1 7 4 6 10 2 11 9 12 3 | 52 | 59 | 56 |
| 17 | 12 2 5 11 10 0 1 6 3 14 8 9 7 4 13 15 | 52 | 62 | 52 |
| 18 | 13 3 2 8 12 0 5 1 11 6 9 15 4 14 7 10 | 52 | 63 | 52 |
| 19 | 7 13 1 4 9 12 8 5 15 14 0 6 11 2 3 10 | 52 | 59 | 52 |
| 20 | 8 11 12 10 2 0 15 1 14 6 4 3 7 9 5 13 | 54 | 61 | 58 |
| 21 | 6 8 12 13 7 2 5 14 9 3 1 15 11 0 10 4 | 54 | 65 | 54 |
| 22 | 9 12 2 5 11 1 10 14 0 4 3 8 6 15 7 13 | 54 | 67 | 60 |
| 23 | 10 12 11 7 8 9 14 5 3 13 4 1 6 0 2 15 | 56 | 69 | 56 |
| 24 | 3 10 14 5 1 12 11 8 15 7 9 6 2 0 13 4 | 56 | 71 | 58 |
| 25 | 9 3 12 5 4 14 6 11 8 7 15 13 10 0 2 1 | 56 | 71 | 60 |
| | **Average** | **48.48** | **58.76** | **50.4** |

Table 5 [9] shows the comparison between the two searches, UA* and BA* with HH, which were implemented in this paper. According to Table 5, the space and time complexity of the UA* algorithm is $O\left(b^d\right)$, where $b$ is the branching and $d$ is the depth of solution, whereas the space and time complexity of the BA* algorithm is $O\left(b^{d/2}\right)$, since two searches are run in the BA* algorithm; thus, the solution depth is divided by two. One significant point to notice is that the time and space complexity of the A* algorithm strongly depends on the heuristics, which heuristics are used, and how they are implemented [9]. Therefore, in this paper, we took advantage of the bidirectional search, the heuristics, and the way of implementing them (as shown in Equation (3)) to reduce the space complexity. Table 5 also presents the completeness and optimality of UA* and BA* with HH. It shows that both the searches are complete but not optimal. This is because of the way of using heuristics, as we mentioned before that BA* with HH guarantees either an optimal or near-optimal solution.

**Table 5.** Evaluation of UA* and BA* searches.

| Criterion | UA* with HH | BA* with HH |
|-----------|-------------|-------------|
| Time complexity | $O\left(b^d\right)$ | $O\left(b^{d/2}\right)$ |
| Space complexity | $O\left(b^d\right)$ | $O\left(b^{d/2}\right)$ |
| Complete | Yes | Yes |
| Optimal | **No** | **No** |

### 5.3. Experiments

In the study, the BA* algorithm was applied by combining the advantages of WD, LC, and MD heuristics. The algorithm was run on the 100 random initial states presented by [8], which is mainly to show the efficiency and performance of our implementation of BA*. In Korf's goal state, the blank is located at the top left corner, and Korf used the IDA* search algorithm with the MD heuristic. Then, those 100 random initial states were used by [14], but this time the IDA* algorithm with MD and LC heuristics was run on them, and the result has been added to Table 6. Even though the implementation of the IDA* algorithm with MD and LC heuristics is quite old, we still compare our results with its results because we also use both MD and LC heuristics, albeit in different ways and with another heuristic (WD). Furthermore, our algorithm considerably reduces the number of generated nodes compared to the results of IDA* with the two heuristics MD and LC.

Table 6 shows that the number of states examined using BA* with HH is much less than the number of states examined using IDA* with MD and LC. For example, the average cost of states examined using IDA* with MD and LC is 37,596,318 states, while the average cost of states generated by BA* with HH is only 48,420 states. Furthermore, Table 6 also shows that the average solution cost that is obtained by BA* with HH is about 55.01, and this is very near the average optimal solution cost which is about 53.1 moves. In addition, the number of moves and the number of generated states in the solution of each instance using both the IDA* and BA* algorithms are also shown in Table 6. Moreover, it is evident in Table 6 that the solution length of 98% of the instances from zero to six moves is far from their optimal solution lengths.

Table 6 also demonstrates the number of state expansions and the WD, MD, and LC values for each of the puzzle instances. Figure 8 graphically shows the total number of states according to the cost difference between their optimal solutions and the solutions achieved by BA* with HH based on Table 6. Table 6 also shows that the solution of 39 states is zero moves away from optimum (they are optimal solutions), the solution of 36 states is two moves away from optimum, the solution of 17 states is four moves away from optimum, and the solution of six states are six moves away from optimum. The figure also shows that the solution of only one state is eight moves away from optimum and the solution of only one state is 10 moves away from optimum. Table 6 also presents the HH value for each instance and it shows that the HH value is very near to the instance's optimal length. The average HH value is 54.04, only 0.97 away from the average optimal length, which is 53.07. The last column of Table 6 indicates the time (in seconds) it takes to solve each puzzle instance by BA* with HH.

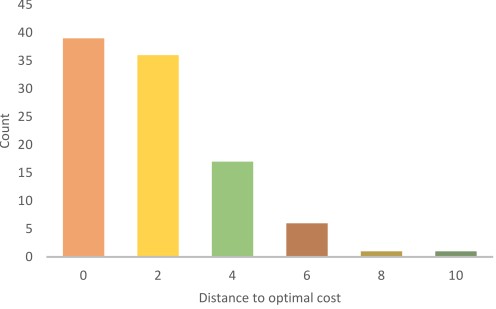

**Figure 8.** Total number of states according to the cost difference between their optimal solutions and the solutions achieved by BA* with HH.

In short, the most important thing about our implementation of BA* with HH is that it drastically reduces the search space without consuming a lot of storage space, since for all three heuristics used in this paper, only 25 KB is required. Furthermore, the results for each instance are very close to the shortest path length, even for complex puzzle states. Additionally, the estimation of HH for each instance shown Table 6 is near to the optimal length.

**Table 6.** Comparison of IDA* algorithm with MD and LC and BA* algorithm with HH for Korf's 100 instances.

| NO | INITIAL STATE | Optimal LEN | IDA* with MD + LC (Generated States) | BA* with HH (Generated States) | LEN (BA*) | BA* with HH (State Expansion) | WD | MD | LC | HH Value | Time (s) (BA*) |
|---|---|---|---|---|---|---|---|---|---|---|---|
| 1 | 14 13 15 7 11 12 9 5 6 0 2 1 4 8 10 3 | 57 | 12,205,623 | 4348 | 59 | 2146 | 43 | 41 | 2 | 59 | 0.15 |
| 2 | 13 5 4 10 9 12 8 14 2 3 7 1 0 15 11 6 | 55 | 4,556,067 | 104,760 | 59 | 50,143 | 45 | 43 | 0 | 59 | 1.16 |
| 3 | 14 7 8 2 13 11 10 4 9 12 5 0 3 6 1 15 | 59 | 156,590,306 | 39,851 | 59 | 18,672 | 43 | 41 | 0 | 57 | 0.59 |
| 4 | 5 12 10 7 15 11 14 0 8 2 1 13 3 4 9 6 | 56 | 9,052,179 | 128,358 | 58 | 62,894 | 44 | 42 | 0 | 58 | 1.22 |
| 5 | 4 7 14 13 10 3 9 12 11 5 6 15 1 2 8 0 | 56 | 2,677,666 | 20,413 | 58 | 9635 | 44 | 42 | 2 | 60 | 0.41 |
| 6 | 14 7 1 9 12 3 6 15 8 11 2 5 10 0 4 13 | 52 | 4,151,682 | 4682 | 54 | 2234 | 40 | 36 | 4 | 56 | 0.15 |
| 7 | 2 11 15 5 13 4 6 7 12 8 10 1 9 3 14 0 | 52 | 97,264,710 | 150,410 | 54 | 75,251 | 34 | 30 | 0 | 44 | 1.57 |
| 8 | 12 11 15 3 8 0 4 2 6 13 9 5 14 1 10 7 | 50 | 3,769,804 | 22,921 | 54 | 11,354 | 36 | 32 | 4 | 51 | 0.43 |
| 9 | 3 14 9 11 5 4 8 2 13 12 6 7 10 1 15 0 | 46 | 88,588 | 1811 | 48 | 871 | 34 | 32 | 4 | 49 | 0.10 |
| 10 | 13 11 8 9 0 15 7 10 4 3 6 14 5 12 2 1 | 59 | 48,531,591 | 42,218 | 59 | 20,030 | 47 | 43 | 2 | 63 | 0.60 |
| 11 | 5 9 13 14 6 3 7 12 10 8 4 0 15 2 11 1 | 57 | 25,537,948 | 67,872 | 59 | 32,265 | 45 | 43 | 2 | 61 | 0.86 |
| 12 | 14 1 9 6 4 8 12 5 7 2 3 0 10 11 13 15 | 45 | 179,628 | 633 | 45 | 298 | 37 | 35 | 0 | 49 | 0.05 |
| 13 | 3 6 5 2 10 0 15 14 1 4 13 12 9 8 11 7 | 46 | 1,051,213 | 14,327 | 48 | 7164 | 36 | 36 | 2 | 50 | 0.34 |
| 14 | 7 6 8 1 11 5 14 10 3 4 9 13 15 2 0 12 | 59 | 53,050,799 | 153,470 | 63 | 76,125 | 43 | 41 | 2 | 59 | 1.47 |
| 15 | 13 11 4 12 1 8 9 15 6 5 14 2 7 3 10 0 | 62 | 130,071,656 | 43,608 | 64 | 20,685 | 46 | 44 | 2 | 63 | 0.54 |
| 16 | 1 3 2 5 10 9 15 6 8 14 13 11 12 4 7 0 | 44 | 2,421,878 | 67,984 | 44 | 34,375 | 24 | 24 | 2 | 34 | 0.86 |
| 17 | 15 14 0 4 11 1 6 13 7 5 8 9 3 2 10 12 | 66 | 100,843,886 | 206,372 | 76 | 98,417 | 54 | 46 | 0 | 69 | 2.38 |
| 18 | 6 0 14 12 1 15 9 10 11 4 7 2 8 3 5 13 | 55 | 5,224,645 | 19,272 | 57 | 9121 | 43 | 43 | 0 | 57 | 0.38 |
| 19 | 7 11 8 3 14 0 6 15 1 4 13 9 5 12 2 10 | 46 | 385,369 | 5381 | 46 | 2475 | 36 | 36 | 2 | 50 | 0.18 |
| 20 | 6 12 11 3 13 7 9 15 2 14 8 10 4 1 5 0 | 52 | 3,642,638 | 32,036 | 54 | 15,414 | 36 | 36 | 0 | 48 | 0.51 |
| 21 | 12 8 14 6 11 4 7 0 5 1 10 15 3 13 9 2 | 54 | 43,980,448 | 59,920 | 56 | 28,403 | 40 | 34 | 2 | 53 | 0.91 |
| 22 | 14 3 9 1 15 8 4 5 11 7 10 13 0 2 12 6 | 59 | 79,549,136 | 4517 | 63 | 2112 | 45 | 41 | 4 | 63 | 0.14 |
| 23 | 10 9 3 11 0 13 2 14 5 6 4 7 8 15 1 12 | 49 | 770,088 | 43,664 | 51 | 21,113 | 37 | 33 | 4 | 52 | 0.64 |
| 24 | 7 3 14 13 4 1 10 8 5 12 9 11 2 15 6 0 | 54 | 15,062,608 | 31,366 | 54 | 14,750 | 38 | 34 | 4 | 53 | 0.66 |
| 25 | 11 4 2 7 1 0 10 15 6 9 14 8 3 13 5 12 | 52 | 13,453,743 | 5258 | 52 | 2485 | 36 | 32 | 4 | 51 | 0.19 |
| 26 | 5 7 3 12 15 13 14 8 0 10 9 6 1 4 2 11 | 58 | 50,000,803 | 110,470 | 58 | 53,289 | 42 | 40 | 4 | 59 | 1.20 |
| 27 | 14 1 8 15 2 6 0 3 9 12 10 13 4 7 5 11 | 53 | 31,152,542 | 37,847 | 55 | 19,212 | 37 | 33 | 2 | 50 | 0.63 |
| 28 | 13 14 6 12 4 5 1 0 9 3 10 2 15 11 8 7 | 52 | 1,584,197 | 16,633 | 54 | 7814 | 40 | 36 | 0 | 52 | 0.39 |
| 29 | 9 8 0 2 15 1 4 14 3 10 7 5 11 13 6 12 | 54 | 10,085,238 | 21,435 | 54 | 10,644 | 42 | 38 | 2 | 57 | 0.44 |
| 30 | 12 15 2 6 1 14 4 8 5 3 7 0 10 13 9 11 | 47 | 680,254 | 21,016 | 47 | 10,296 | 35 | 35 | 0 | 47 | 0.41 |

**Table 6.** *Cont.*

| NO | INITIAL STATE | Optimal LEN | IDA* with MD + LC (Generated States) | BA* with HH (Generated States) | LEN (BA*) | BA* with HH (State Expansion) | WD | MD | LC | HH Value | Time (s) (BA*) |
|----|---------------|-------------|-------------------------------------|------------------------------|-----------|-------------------------------|----|----|----|----------|----------------|
| 31 | 12 8 15 13 1 0 5 4 6 3 2 11 9 7 14 10 | 50 | 538,886 | 514 | 52 | 239 | 40 | 38 | 2 | 55 | 0.05 |
| 32 | 14 10 9 4 13 6 5 8 2 12 7 0 1 3 11 15 | 59 | 183,341,087 | 123,812 | 61 | 58,715 | 43 | 43 | 2 | 59 | 1.23 |
| 33 | 14 3 5 15 11 6 13 9 0 10 2 12 4 1 7 8 | 60 | 28,644,837 | 37,806 | 62 | 17,962 | 44 | 42 | 0 | 58 | 0.61 |
| 34 | 6 11 7 8 13 2 5 4 1 10 3 9 14 0 12 15 | 52 | 1,174,414 | 10,257 | 52 | 4916 | 38 | 36 | 6 | 56 | 0.26 |
| 35 | 1 6 12 14 3 2 15 8 4 5 13 9 0 7 11 10 | 55 | 9,214,047 | 58,967 | 55 | 28,765 | 41 | 39 | 0 | 54 | 0.72 |
| 36 | 12 6 0 4 7 3 15 1 13 9 8 11 2 14 5 10 | 52 | 4,657,636 | 13,346 | 52 | 6485 | 38 | 36 | 2 | 52 | 0.31 |
| 37 | 8 1 7 12 11 0 10 5 9 15 6 13 14 2 3 4 | 58 | 21,274,607 | 29,195 | 58 | 14,151 | 44 | 40 | 2 | 59 | 0.52 |
| 38 | 7 15 8 2 13 6 3 12 11 0 4 10 9 5 1 14 | 53 | 4,946,981 | 2105 | 53 | 1003 | 41 | 41 | 2 | 57 | 0.09 |
| 39 | 9 0 4 10 1 14 15 3 12 6 5 7 11 13 8 2 | 49 | 3,911,623 | 18,877 | 49 | 9056 | 35 | 35 | 0 | 47 | 0.35 |
| 40 | 11 5 1 14 4 12 10 0 2 7 13 3 9 15 6 8 | 54 | 13,107,557 | 120,964 | 56 | 58,445 | 38 | 36 | 2 | 52 | 1.17 |
| 41 | 8 13 10 9 11 3 15 6 0 1 2 14 12 5 4 7 | 54 | 12,388,516 | 5793 | 54 | 2776 | 42 | 36 | 4 | 58 | 0.17 |
| 42 | 4 5 7 2 9 14 12 13 0 3 6 11 8 1 15 10 | 42 | 217,288 | 17,756 | 46 | 8504 | 32 | 30 | 2 | 44 | 0.38 |
| 43 | 11 15 14 13 1 9 10 4 3 6 2 12 7 5 8 0 | 64 | 7,034,879 | 9938 | 68 | 4754 | 54 | 48 | 6 | 76 | 0.21 |
| 44 | 12 9 0 6 8 3 5 14 2 4 11 7 10 1 15 13 | 50 | 3,819,541 | 2461 | 50 | 1244 | 34 | 32 | 6 | 51 | 0.10 |
| 45 | 3 14 9 7 12 15 0 4 1 8 5 6 11 10 2 13 | 51 | 764,473 | 654 | 51 | 294 | 39 | 39 | 0 | 52 | 0.05 |
| 46 | 8 4 6 1 14 12 2 15 13 10 9 5 3 7 0 11 | 49 | 1,510,387 | 4417 | 51 | 2175 | 35 | 35 | 6 | 53 | 0.18 |
| 47 | 6 10 1 14 15 8 3 5 13 0 2 7 4 9 11 12 | 47 | 221,531 | 1173 | 47 | 565 | 35 | 35 | 0 | 47 | 0.08 |
| 48 | 8 11 4 6 7 3 10 9 2 12 15 13 0 1 5 14 | 49 | 255,047 | 2302 | 49 | 1080 | 41 | 39 | 0 | 54 | 0.11 |
| 49 | 10 0 2 4 5 1 6 12 11 13 9 7 15 3 14 8 | 59 | 203,873,877 | 156,955 | 65 | 75,800 | 39 | 33 | 4 | 54 | 1.54 |
| 50 | 12 5 13 11 2 10 0 9 7 8 4 3 14 6 15 1 | 53 | 6,225,180 | 37,831 | 57 | 18,100 | 41 | 39 | 2 | 56 | 0.61 |
| 51 | 10 2 8 4 15 0 1 14 11 13 3 6 9 7 5 12 | 56 | 4,683,054 | 25,419 | 56 | 12,338 | 44 | 44 | 0 | 59 | 0.48 |
| 52 | 10 8 0 12 3 7 6 2 1 14 4 11 15 13 9 5 | 56 | 33,691,153 | 120,510 | 60 | 60,031 | 40 | 38 | 4 | 57 | 1.22 |
| 53 | 14 9 12 13 15 4 8 10 0 2 1 7 3 11 5 6 | 64 | 125,641,730 | 103,879 | 68 | 49,722 | 54 | 50 | 0 | 71 | 1.11 |
| 54 | 12 11 0 8 10 2 13 15 5 4 7 3 6 9 14 1 | 56 | 26,080,659 | 47,294 | 58 | 22,908 | 42 | 40 | 2 | 57 | 0.76 |
| 55 | 13 8 14 3 9 1 0 7 15 5 4 10 12 2 6 11 | 41 | 163,077 | 5291 | 43 | 2400 | 33 | 29 | 2 | 45 | 0.18 |
| 56 | 3 15 2 5 11 6 4 7 12 9 1 0 13 14 10 8 | 55 | 166,183,825 | 153,475 | 59 | 75,038 | 35 | 29 | 4 | 49 | 1.50 |
| 57 | 5 11 6 9 4 13 12 0 8 2 15 10 1 7 3 14 | 50 | 3,977,809 | 8430 | 50 | 4021 | 36 | 36 | 0 | 48 | 0.22 |
| 58 | 5 0 15 8 4 6 1 14 10 11 3 9 7 12 2 13 | 51 | 3,563,941 | 8020 | 51 | 3771 | 39 | 37 | 4 | 55 | 0.21 |
| 59 | 15 14 6 7 10 1 0 11 12 8 4 9 2 5 13 3 | 57 | 90,973,287 | 36,373 | 57 | 17,423 | 39 | 35 | 4 | 55 | 0.65 |
| 60 | 11 14 13 1 2 3 12 4 15 7 9 5 10 6 8 0 | 66 | 256,537,528 | 167,180 | 72 | 80,902 | 54 | 48 | 0 | 70 | 1.69 |

**Table 6.** *Cont.*

| NO | INITIAL STATE | Optimal LEN | IDA* with MD + LC (Generated States) | BA* with HH (Generated States) | LEN (BA*) | BA* with HH (State Expansion) | WD | MD | LC | HH Value | Time (s) (BA*) |
|---|---|---|---|---|---|---|---|---|---|---|---|
| 61 | 6 13 3 2 11 9 5 10 1 7 12 14 8 4 0 15 | 45 | 672,959 | 3024 | 45 | 1471 | 31 | 31 | 4 | 45 | 0.12 |
| 62 | 4 6 12 0 14 2 9 13 11 8 3 15 7 10 1 5 | 57 | 8,463,998 | 23,726 | 61 | 11,426 | 45 | 43 | 2 | 61 | 0.40 |
| 63 | 8 10 9 11 14 1 7 15 13 4 0 12 6 2 5 3 | 56 | 20,999,336 | 14,771 | 56 | 7196 | 42 | 40 | 4 | 59 | 0.34 |
| 64 | 5 2 14 0 7 8 6 3 11 12 13 15 4 10 9 1 | 51 | 43,522,756 | 80,791 | 53 | 38,143 | 37 | 31 | 4 | 51 | 0.96 |
| 65 | 7 8 3 2 10 12 4 6 11 13 5 15 0 1 9 14 | 47 | 2,444,273 | 9450 | 47 | 4669 | 33 | 31 | 4 | 47 | 0.26 |
| 66 | 11 6 14 12 3 5 1 15 8 0 10 13 9 7 4 2 | 61 | 394,246,898 | 57,527 | 61 | 27,714 | 43 | 41 | 2 | 59 | 0.84 |
| 67 | 7 1 2 4 8 3 6 11 10 15 0 5 14 12 13 9 | 50 | 47,499,462 | 154,127 | 56 | 75,339 | 30 | 28 | 2 | 41 | 1.59 |
| 68 | 7 3 1 13 12 10 5 2 8 0 6 11 14 15 4 9 | 51 | 6,959,507 | 28,456 | 51 | 13,873 | 33 | 31 | 4 | 47 | 0.52 |
| 69 | 6 0 5 15 1 14 4 9 2 13 8 10 11 12 7 3 | 53 | 5,186,587 | 48,211 | 57 | 23,657 | 37 | 37 | 2 | 51 | 0.75 |
| 70 | 15 1 3 12 4 0 6 5 2 8 14 9 13 10 7 11 | 52 | 40,161,673 | 85,108 | 52 | 41,672 | 36 | 30 | 2 | 48 | 0.97 |
| 71 | 5 7 0 11 12 1 9 10 15 6 2 3 8 4 13 14 | 44 | 539,387 | 12,680 | 46 | 6422 | 30 | 30 | 4 | 44 | 0.30 |
| 72 | 12 15 11 10 4 5 14 0 13 7 1 2 9 8 3 6 | 56 | 55,514,360 | 147,629 | 64 | 75,073 | 42 | 38 | 2 | 57 | 1.42 |
| 73 | 6 14 10 5 15 8 7 1 3 4 2 0 12 9 11 13 | 49 | 1,130,807 | 1645 | 53 | 809 | 41 | 37 | 2 | 55 | 0.09 |
| 74 | 14 13 4 11 15 8 6 9 0 7 3 1 2 10 12 5 | 56 | 310,312 | 32,986 | 62 | 15,904 | 48 | 46 | 0 | 63 | 0.67 |
| 75 | 14 4 0 10 6 5 1 3 9 2 13 15 12 7 8 11 | 48 | 5,796,660 | 150,985 | 50 | 75,069 | 30 | 30 | 4 | 44 | 1.75 |
| 76 | 15 10 8 3 0 6 9 5 1 14 13 11 7 2 12 4 | 57 | 25,481,596 | 51,179 | 57 | 24,049 | 45 | 41 | 2 | 61 | 0.80 |
| 77 | 0 13 2 4 12 14 6 9 15 1 10 3 11 5 8 7 | 54 | 5,479,397 | 62,726 | 58 | 30,141 | 42 | 34 | 2 | 55 | 0.94 |
| 78 | 3 14 13 6 4 15 8 9 5 12 10 0 2 7 1 11 | 53 | 2,722,095 | 8781 | 55 | 4147 | 43 | 41 | 0 | 57 | 0.22 |
| 79 | 0 1 9 7 11 13 5 3 14 12 4 2 8 6 10 15 | 42 | 107,088 | 4554 | 42 | 2197 | 30 | 28 | 2 | 41 | 0.16 |
| 80 | 11 0 15 8 13 12 3 5 10 1 4 6 14 9 7 2 | 57 | 39,801,475 | 22,413 | 61 | 10,698 | 45 | 43 | 0 | 59 | 0.55 |
| 81 | 13 0 9 12 11 6 3 5 15 8 1 10 4 14 2 7 | 53 | 1,088,123 | 1420 | 53 | 689 | 41 | 39 | 2 | 56 | 0.08 |
| 82 | 14 10 2 1 13 9 8 11 7 3 6 12 15 5 4 0 | 62 | 203,606,265 | 173,460 | 68 | 87,034 | 44 | 40 | 4 | 61 | 1.97 |
| 83 | 12 3 9 1 4 5 10 2 6 11 15 0 14 7 13 8 | 49 | 2,155,880 | 32,271 | 51 | 16,376 | 35 | 31 | 6 | 51 | 0.58 |
| 84 | 15 8 10 7 0 12 14 1 5 9 6 3 13 11 4 2 | 55 | 17,323,672 | 100,981 | 57 | 49,825 | 39 | 37 | 6 | 57 | 1.14 |

**Table 6.** *Cont.*

| NO | INITIAL STATE | Optimal LEN | IDA* with MD + LC (Generated States) | BA* with HH (Generated States) | LEN (BA*) | BA* with HH (State Expansion) | WD | MD | LC | HH Value | Time (s) (BA*) |
|---|---|---|---|---|---|---|---|---|---|---|---|
| 85 | 4 7 13 10 1 2 9 6 12 8 14 5 3 0 11 15 | 44 | 933,953 | 11,604 | 46 | 5594 | 32 | 32 | 0 | 43 | 0.31 |
| 86 | 6 0 5 10 11 12 9 2 1 7 4 3 14 8 13 15 | 45 | 237,466 | 4906 | 47 | 2342 | 35 | 35 | 2 | 49 | 0.17 |
| 87 | 9 5 11 10 13 0 2 1 8 6 14 12 4 7 3 15 | 52 | 7,928,514 | 38,524 | 52 | 19,390 | 36 | 34 | 2 | 49 | 0.59 |
| 88 | 15 2 12 11 14 13 9 5 1 3 8 7 0 10 6 4 | 65 | 422,768,851 | 85,817 | 67 | 42,265 | 49 | 43 | 2 | 65 | 1.14 |
| 89 | 11 1 7 4 10 13 3 8 9 14 0 15 6 5 2 12 | 54 | 29,171,607 | 50,303 | 54 | 23,800 | 40 | 38 | 2 | 55 | 0.71 |
| 90 | 5 4 7 1 11 12 14 15 10 13 8 6 2 0 9 3 | 50 | 649,591 | 15,343 | 52 | 7592 | 36 | 36 | 4 | 52 | 0.36 |
| 91 | 9 7 5 2 14 15 12 10 11 3 6 1 8 13 0 4 | 57 | 91,220,187 | 36,250 | 57 | 17,644 | 43 | 41 | 0 | 57 | 0.56 |
| 92 | 3 2 7 9 0 15 12 4 6 11 5 14 8 13 10 1 | 57 | 68,307,452 | 35,707 | 57 | 17,553 | 39 | 37 | 2 | 53 | 0.65 |
| 93 | 13 9 14 6 12 8 1 2 3 4 0 7 5 10 11 15 | 46 | 350,208 | 72,971 | 50 | 35,114 | 36 | 34 | 0 | 47 | 0.95 |
| 94 | 5 7 11 8 0 14 9 13 10 12 3 15 6 1 4 2 | 53 | 390,368 | 4655 | 59 | 2158 | 45 | 45 | 0 | 60 | 0.15 |
| 95 | 4 3 6 13 7 15 9 0 10 5 8 11 2 12 1 14 | 50 | 1,517,920 | 14,900 | 54 | 6986 | 42 | 34 | 2 | 55 | 0.39 |
| 96 | 1 7 15 14 2 6 4 9 12 11 13 3 0 8 5 10 | 49 | 1,157,734 | 9322 | 51 | 4642 | 37 | 35 | 2 | 51 | 0.26 |
| 97 | 9 14 5 7 8 15 1 2 10 4 13 6 12 0 11 3 | 44 | 166,566 | 7933 | 44 | 3829 | 32 | 32 | 2 | 45 | 0.24 |
| 98 | 0 11 3 12 5 2 1 9 8 10 14 15 7 4 13 6 | 54 | 41,564,669 | 72,441 | 56 | 35,008 | 38 | 34 | 0 | 49 | 0.95 |
| 99 | 7 15 4 0 10 9 2 5 12 11 13 6 1 3 14 8 | 57 | 18,038,550 | 145,912 | 59 | 69,971 | 43 | 39 | 0 | 56 | 1.52 |
| 100 | 11 4 0 8 6 10 5 13 12 7 14 3 1 2 9 15 | 54 | 17,778,222 | 112,634 | 56 | 53,227 | 40 | 38 | 2 | 55 | 1.35 |
| | **SUM** | **5307** | **3,759,631,814** | **4,841,970** | **5501** | **2,353,978** | **3957** | **3705** | **212** | **5404** | **64** |
| | **Average** | **53.07** | **37,596,318** | **48,420** | **55.01** | **23,540** | **40** | **37** | **2** | **54.04** | **0.64** |

The state-of-the-art heuristics that have been designed for solving the Fifteen Puzzle instances are based on machine learning or deep learning techniques [56–58]. They have achieved good results, but these algorithms used a training dataset that contains millions of difficult Fifteen Puzzle states with their optimal solution costs, which were calculated using the IDA* [56] algorithm with the 7–8 PDB heuristic. As we mentioned before, 7–8 PDB is the most efficient heuristic for solving the Fifteen Puzzle problems, but this heuristic requires a huge amount of space for storing the entries (57,657,600 entries for seven tiles and 518,918,400 entries for eight tiles) [59], which takes up 575 MB [18] to 4.5 GBs [56] of memory depending on the techniques which have been used for storing the entries. Moreover, the training dataset needed by these heuristics, also requires a lot of space. Additionally, these heuristics may not achieve a good result for every Fifteen Puzzle state, since they heavily depend on the size and proportion of the training dataset, and there are about $10^{13}$ solvable Fifteen Puzzle states, which is an enormous figure. Another weakness of these heuristics is the large amount of time that they require to process [56,57], which mainly depends on the size of the training dataset. On the other hand, the heuristics used in this study are those which can be calculated during the search and each puzzle state is taken into account. Additionally, they do not need databases or tables to store the precomputation results (except 25 KB, which is only used for calculating the WD heuristic value for the sake of speeding up the search). To the best our knowledge, the approach we have used has not been used for a long time in research, since it generates a lot of states, which causes the A* algorithm to run out of memory since it needs to keep all the generated nodes in memory, and IDA* needs hours to solve most of the Fifteen Puzzle problems. Here, we have used these heuristics in a way that can significantly reduce the generated states without exhausting the memory or taking a long time to solve the Fifteen Puzzle problems. In spite of this, in Table 7, we still compare our results with a focal discrepancy search (FDS) and focal search (FS). These two searches use a learned heuristic DeepCubeA, which was trained over 1.5 million Fifteen Puzzle problems, and for the evaluation, Korf's 100 instances were used [57,58]. Table 7 shows that only BA* with HH and FDS can solve the 100 instances. The FDS and FS expanded fewer nodes compared to the results of BA* with HH. The average solution cost that is obtained by BA* with HH is nearer to the average optimal cost, which is 53.1 moves, than that of FDS. FS obtained the best average cost, but this search is only able to solve 83% of the puzzle instances. As mentioned before, one of the weaknesses of the heuristics that depend on the dataset is speed, as Table 7 shows that, in terms of runtime, our proposed algorithm significantly outperforms both FDS and FS. We believe that our proposed algorithm has obtained a good result, since we have used heuristics that can be calculated during the search without requiring a huge amount of space to store precomputation results. Moreover, the approach we have used expands an average of 23,540 nodes, which is small, while in the past it expanded an average of millions of nodes.

**Table 7.** Results for the Fifteen Puzzle with the algorithms FS, FDS, and BA* with HH.

|  | Coverage | Expansions (avg) | Cost (avg) | Average Time (seconds) |
|---|---|---|---|---|
| BA* with HH | 100% | 23,540 | 55.01 | 0.64 |
| FS (h nn) | 83% | 10,414 | 54.57 | >100 |
| FDS (best) | 100% | 1478 | 55.47 | >10 |
| FDS (rank) | 100% | 6542 | 55.45 | >10 |

## 6. Conclusions

In this paper, we proposed a bidirectional A* (BA*) search algorithm with three heuristics, WD, LC, and MD, where the heuristics are combined in a way that guides the algorithm efficiently toward the solution and expands fewer states. It is clear that our implementation of the BA* algorithm does not find the optimal solution for most of the Fifteen Puzzle problem instances, but the solutions are very close to optimal length.

Additionally, we proved, using empirical evidence, that the BA* heuristic search algorithm is more efficient than the UA* heuristic search algorithm in terms of state expansions.

Accordingly, designing a heuristic function to accurately choose the next state while exploring the space is challenging due to the huge size of the Fifteen Puzzle; it has $10^{13}$ states. To evaluate the performance and efficiency of HH with the BA* algorithm, we made some comparisons, especially in terms of optimality and space complexity. We showed that HH with the BA* algorithm produces acceptable results and hugely reduces the search space.

In future work, hybridizing heuristic (HH) should be used to increase the effectiveness of metaheuristic algorithms in solving the Fifteen Puzzle, since HH requires a very small amount of space, and it is effective for estimating the complexity of puzzle problems. Therefore, we recommend using the novel metaheuristic algorithms such as FDO [60], LPB [61], and ANA [62] for the Fifteen Puzzle instead of the ABC algorithm, since these metaheuristic algorithms work toward optimality.

However, there are some limitations of the algorithm proposed in this paper that need to be considered. First, overestimating the actual path cost from the current state to the goal state for some of the Fifteen Puzzle instances, as s shown in the last column in Table 6, leads to non-optimal solutions. Future work can focus on reducing the overestimation, which is somewhat small. Second, all the states generated by the proposed algorithm while traversing are stored in memory and some of them are highly likely not to be expanded. It would be helpful to specify those states that will never be expanded and discard them to reduce memory consumption. Third, in this paper, we have not tried to speed up the heuristics (except for WD, but there can be other ways to further increase the speed of this heuristic calculation value). However, each one of the heuristics that is used in this paper can be speeded up. In future work, to decrease the time complexity, the number of generated nodes and the time taken to calculate the heuristic per node must be reduced.

**Author Contributions:** Conceptualization, D.O.H., A.M.A. and T.A.R.; methodology, D.O.H., A.M.A. and H.S.T.; software, D.O.H.; validation, H.S.T.; formal analysis, T.A.R.; investigation, S.M.; data curation, A.M.A.; writing – original draft, D.O.H.; writing – review & editing, D.O.H., T.A.R. and S.M. All authors have read and agreed to the published version of the manuscript.

**Funding:** This research received no external funding.

**Institutional Review Board Statement:** Not applicable.

**Informed Consent Statement:** Not applicable.

**Data Availability Statement:** All data used are included in this published article.

**Conflicts of Interest:** The authors declare no conflict of interest.

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
