# Peer review of "The Fifteen Puzzle—A New Approach through Hybridizing Three Heuristics Methods"

_computers, doi:10.3390/computers12010011_

Round 1

Reviewer 1 Report

This work proposes a Bidirectional A* (BA*) search algorithm with three heuristics WD, LC, and MD, and the heuristics are combined in a way that guides the algorithm efficiently toward the solution and expands fewer states. The algorithm was run on the 100 random initial states.

Some more detailed comments are given below. I hope that if the authors will take them into account the paper will be improved.

1. There are many journal paper concerning the variants of A* search algorithm from the last two years. 

Authors should survey and discuss these state-of-the-art methods in Section 2.

(1) Optimal path planning with modified A-Star algorithm for stealth unmanned aerial vehicles in 3D network radar environment, 2022.

(2) An Efficient and Robust Improved A* Algorithm for Path Planning, 2021.

(3) A D2D Group Communication Scheme Using Bidirectional and InCremental A-Star Search to Configure Paths, 2022.

(4) Global path planning based on a bidirectional alternating search A* algorithm for mobile robots, 2022.

2. It is hard to link the proposed algorithm I (BA* algorithm) with the test examples.

A simple numerical example is needed to provide an illustration of the BA* algorithm in Section 2.

3. Except the IDA* algorithm, authors should compare the performance of the proposed method with other existing state-of-the-art methods for solving Fifteen Puzzle in Section 5.

Author Response

Response Letter- Reviewer No.1

Article Title: The Fifteen Puzzle- A New Approach through Hybridizing Three Heuristics Methods

Author’s: Dler O. Hasan et al

Thank you for taking the time to leave your judgments. We appreciate the time and effort you put into your comments on our manuscript Article on behalf of my co-authors. We appreciate the careful review and constructive suggestions and your feedback is really beneficial. We carefully considered the feedback and made significant changes to the manuscript. It is our belief that the manuscript is substantially improved after making the suggested edits.

Our replies are listed below in a point-by-point format. We've provided a corrected version of our manuscript, with all of the revisions highlighted in yellow color.

Part #1:

There are many journal paper concerning the variants of A* search algorithm from the last two years. 

Authors should survey and discuss these state-of-the-art methods in Section 2.

(1) Optimal path planning with modified A-Star algorithm for stealth unmanned aerial vehicles in 3D network radar environment, 2022.

(2) An Efficient and Robust Improved A* Algorithm for Path Planning, 2021.

(3) A D2D Group Communication Scheme Using Bidirectional and InCremental A-Star Search to Configure Paths, 2022.

(4) Global path planning based on a bidirectional alternating search A* algorithm for mobile robots, 2022.

  • Response: We surveyed and discussed those methods in Section 2 (Please see P3 L112-120 for detailed information)

Part #2:

It is hard to link the proposed algorithm I (BA* algorithm) with the test examples.

A simple numerical example is needed to provide an illustration of the BA* algorithm in Section 2.

  • Response: Thanks for the valuable recommendation; to link our proposed algorithm with the test examples, we provided a numerical example with a figure (Please see P4-5 L185-198 and Figure 1).

During the preparation of the numerical example, we discovered that there is a mistake with calculating the LC heuristic only for the goal state with a blank at the top left corner. This occurred because we mistakenly used the same method, we created for calculating the LC heuristic for the goal state with a blank at the bottom right corner. Fortunately, after solving that problem, the discussion and conclusion of the paper remained unchanged. However, it led to improving the efficiency of the algorithm. Therefore, the only things that changed in the manuscript were the numbers that we highlighted in turquoise color. We are sorry about that.

Part #3:

Except the IDA* algorithm, authors should compare the performance of the proposed method with other existing state-of-the-art methods for solving Fifteen Puzzle in Section 5.

  • Response: We compared the performance of our proposed algorithm with other state-of-the-art methods and we provided some reasons that our approach is more appropriate and efficient. (Please see P19-20 L586-623 and Table 7 for detailed information)

We also add a new column name “Time” to Table 6 which presents the time it takes to solve each puzzle instance.

Once again, thank you very much for your comments and suggestions.

Note: we have modified several notes according to other reviewer’s suggestions and comments.

Reviewer 2 Report

This paper proposes an algorithm for solving the fifteen puzzle problems. The paper is well written and explained. I only have a few comments and suggestions:
1. In the experiment section, the authors only compare the proposed algorithm with other conventional algorithms. In this experiment, the authors used the same 100 random initial states presented in [8]. There are some previous studies that have also proposed algorithms for solving the fifteen puzzle problems. Thus, I would like to suggest that the authors compare the proposed algorithm's performance with previous studies. Using the same initial state, I think it is possible to compare the proposed study with previous studies.

2. Once the proposed algorithm has an advantage or superiority, it might have limitations. Thus, please highlight the limitation in the conclusion.

Author Response

Response Letter- Reviewer No.2

Article Title: The Fifteen Puzzle- A New Approach through Hybridizing Three Heuristics Methods

Author: Dler O. Hasan et al

Thank you for taking the time to leave your judgments. We appreciate the time and effort you put into your comments on our manuscript Article on behalf of my co-authors. We appreciate the careful review and constructive suggestions and your feedback is really beneficial. We carefully considered the feedback and made significant changes to the manuscript. It is our belief that the manuscript is substantially improved after making the suggested edits.

Our replies are listed below in a point-by-point format. We've provided a corrected version of our manuscript, with all of the revisions highlighted in yellow color.

Part #1:

In the experiment section, the authors only compare the proposed algorithm with other conventional algorithms. In this experiment, the authors used the same 100 random initial states presented in [8]. There are some previous studies that have also proposed algorithms for solving the fifteen puzzle problems. Thus, I would like to suggest that the authors compare the proposed algorithm's performance with previous studies. Using the same initial state, I think it is possible to compare the proposed study with previous studies.

  • Response: We compared the performance of our proposed algorithm with other state-of-the-art methods and we provided some reasons that our approach is more appropriate and efficient. (Please see P19-20 L586-623 and Table 7 for detailed information)

We also add a new column name “Time” to Table 6 which presents the time it takes to solve each puzzle instance.

Part #2:

Once the proposed algorithm has an advantage or superiority, it might have limitations. Thus, please highlight the limitation in the conclusion.

  • Thanks for your constructive comment and it is really helpful for our research. We identified the limitations of our study (Please see P20 L645-657 for detailed information)

During the preparation of a numerical example, we discovered that there is a mistake with calculating the LC heuristic only for the goal state with a blank at the top left corner. This occurred because we mistakenly used the same method, we created for calculating the LC heuristic for the goal state with a blank at the bottom right corner. Fortunately, after solving that problem, the discussion and conclusion of the paper remained unchanged. However, it led to improving the efficiency of the algorithm. Therefore, the only things that changed in the manuscript were the numbers that we highlighted in turquoise color. We are sorry about that.

Once again, thank you very much for your comments and suggestions.

Note: we have modified several notes according to other reviewers’ suggestions and comments.

Round 2

Reviewer 1 Report

The authors have carefully addressed the previous comments of the reviewer and significantly improved the manuscript. 

Reviewer 2 Report

All the suggestions have been addressed.